# Spawning-Induced pH Increase Activates Sperm Attraction and Fertilization Abilities in Eggs of the Ascidian, *Phallusia philippinensis* and *Ciona intestinalis*

**DOI:** 10.3390/ijms24032666

**Published:** 2023-01-31

**Authors:** Noburu Sensui, Yosinori Itoh, Nobuhiko Okura, Kogiku Shiba, Shoji A. Baba, Kazuo Inaba, Manabu Yoshida

**Affiliations:** 1Department of Human Biology and Anatomy, Faculty of Medicine, University of the Ryukyus, Nishihara 903-0213, Japan; 2Department of Molecular Anatomy, Faculty of Medicine, University of the Ryukyus, Nishihara 903-0213, Japan; 3Shimoda Marine Research Center, University of Tsukuba, Shimoda 415-0025, Japan; 4Department of Biology, Graduate School of Humanities and Sciences, Ochanomizu University, Tokyo 112-8610, Japan; 5Misaki Marine Biological Station, School of Science, The University of Tokyo, Miura 238-0225, Japan

**Keywords:** ascidian, fertilizing ability, sperm chemotaxis, oocyte maturation

## Abstract

In Phlebobranchiata ascidians, oocytes and spermatozoa are stored in the oviduct and spermiduct, respectively, until spawning occurs. Gametes in the gonoducts are mature and fertilizable; however, it was found that the gametes of the ascidians *Phallusia philippinensis* and *Ciona intestinalis* could not undergo fertilization in the gonoductal fluids. The body fluids of the ascidians, especially in the gonoducts, were much more acidic (pH 5.5–6.8) than seawater (pH 8.2), and the fertilization rate was low under such acidic conditions. Hence, we examined the effect of pH on gametes. Pre-incubation of gonoductal eggs at pH 8.2 prior to insemination increased fertilization rates, even when insemination was performed under low pH conditions. Furthermore, an increase in ambient pH induced an increase in the intracellular pH of the eggs. It was also found that an increase in ambient pH triggered the release of sperm attractants from the egg and is therefore necessary for sperm chemotaxis. Hence, acidic conditions in the gonoductal fluids keep the gametes, especially eggs, infertile, and the release of eggs into seawater upon spawning induces an increase in ambient pH, which enables egg fertilization.

## 1. Introduction

Upon release from the ovary or testis, the gametes are transported to the site of fertilization. During this transportation, the gametes are affected by various environmental factors. It is known that the fertilization ability of mature gametes changes depending on the external environment. In particular, sperm functions, such as initiation and activation of motility, chemotaxis to the egg, and acrosome reaction, are regulated by various external factors [1]. Therefore, to fully understand the process of fertilization, it is necessary to examine how gamete fertility is regulated by the external environment.

In spermatozoa, changes in the intracellular pH (pH_i_) induced by extracellular factors regulate sperm motility and fertility [2,3]. In species with internal fertilization, such as mammals, spermatozoa acquire their fertilizing ability, called capacitation, in the female reproductive tract. This ability is acquired through the sequential effects of the surrounding environmental factors, such as hormones, ions, and other chemicals [4]. In the female reproductive tracts of mammals, the pH in the intravaginal area is kept low (pH < 5 in human [5]; pH 7.2 in rodents [6]), and from the uterus to the fallopian tubes, the pH is 7–8 [7]. Thus, the spermatozoa face an increase in pH during the process of progressing through the female reproductive tract, and this increase in extracellular pH is thought to result in alkalinization within the sperm, which regulates sperm motility and fertility [8]. The motility and fertilization of human spermatozoa are known to decrease in media below pH 6.2 [9]. In species with external fertilization, gametes are released from the body into the outer environment, where they are subjected to the influence of external factors, including changes in pH. Especially in marine animals, body fluids generally have a pH of 7.2–7.4; however, seawater has a pH of approximately 8.2, which is much higher than that in the body. The sperm of sea urchins in the male body are immotile because of the low-pH environment, and an increase in pH at the time of spawning is considered to trigger sperm motility [10]. In salmonids and barbel, which are fertilized in freshwater, an elevated pH due to ovarian fluid is also involved in the fertilization potential of eggs and sperm motility [2,11,12].

In recent decades, the molecular mechanisms involved in the regulatory effects of pH increase on the sperm function have been elucidated. The sperm-specific Ca^2+^ channel CatSper, which is the only functioning Ca^2+^ channel in mammalian sperm, opens upon a pH increase [13,14,15]. An increase in pH_i_ is sensed by EFCAB9, a subunit of CatSper, leading to a change in its interaction with another subunit, CatSperζ, which is involved in the regulation of channel gating [15]. The motility and chemotactic behavior of echinoderm spermatozoa are also mediated by CatSper [16]. The K^+^ channel Slo3 is another sperm-specific channel sensitive to pH_i_, which is thought to regulate the sperm function through changes in pH and membrane potential [3,17].

In eggs, the structure and progression of meiosis appear to be mediated by extracellular factors, including pH changes. The ultrastructure of the zona pellucida of mammalian oocytes changes during ovulation. This process is associated with the exposure to oviductal fluid, and it results in the zona’s ability to block polyspermy [18,19]. In starfish oocytes, meiotic division is arrested at the first meiotic metaphase (MI) stage, and meiosis restarts when the oocytes are released into seawater, which has a higher pH than the gonoductal fluids [20,21,22]. However, little is known about the effects of the surrounding environment, including ambient pH, on egg fertility in other species. Furthermore, the detailed pH_i_-regulated signaling pathways in oocytes are still not well understood.

Ascidians are marine invertebrates belonging to the primitive chordates. Especially *Ciona intestinalis*, due to its a small genome, its whole genome was elucidated in 2002 [23], and now it has been one of the most used marine invertebrates as a model organism for fertilization and developmental studies in recent years. The oocytes of Phlebobranchiata ascidians are released from the ovary into the oviduct after the germinal vesicle breakdown and arrested at the MI stage as unfertilized eggs. The unfertilized eggs are stored in the oviduct until spawning. Spermatozoa that have completed spermiogenesis in the testis are transferred to, and stored in, the spermiduct. The stored oocytes and spermatozoa in these gonoducts are considered mature and fertile because they can undergo fertilization in seawater. However, it is currently not known how environmental factors affect gamete fertility. In this study, we focused on the effects of pH, which changes significantly upon spawning, on gamete fertility of ascidians. The results revealed that an increase in extracellular pH is vital for triggering the final changes that enable fertilization in ascidian gametes.

## 2. Results

### 2.1. The pH in Different Ascidian Organs and Fluids Is Lower Than That of Sea Water

Intraluminal pH was first measured by direct insertion of a pH electrode into different organs of *Ciona intestinalis* and *Phallusia philippinensis*. In the heart of *C. intestinalis*, the body fluid pH was 6.8 ± 0.1, which is lower than the pH of seawater in Japan (pH 8.0–8.2). The pH in the ovary, oviduct, and spermiduct was between 6.4 and 6.7, which is similar to or slightly lower than that in the heart of *C. intestinalis* (Table 1). In the oviducts and spermiducts of *P. philippinensis*, the pH was between 5.5 and 6.8, which is also lower than that of seawater (Table 1). The pH was also measured in the body fluids collected from the organs, which were also acidic. Fluids from the heart (body fluid), oviducts (ovarian fluid), and spermiducts (seminal fluid) had a pH range of 6.0–6.8 in both *Phallusia* and *Ciona* (Table 1). This indicates that the ascidian body, including the reproductive organs, is an acidic environment, and gametes are preserved under low pH conditions. The pH of the perivitelline space in *Phallusia* eggs was measured using phenol red. The value was almost identical to the pH outside the vitelline coat, indicating that the low-pH environment in the ovary and oviduct directly results in the low pH condition of the egg.

### 2.2. Low pH Inhibits Fertilization in Ascidians

As described above, ascidian spermatozoa and oocytes are stored under acidic conditions prior to spawning. Furthermore, the gametes of both ascidians could not fertilize in the gonoductal fluids when eggs were experimentally inseminated in vitro. No fertilized eggs were observed within 3 h of insemination, when almost the same volume of oocytes and semen were taken from the gonoducts of different individuals and mixed (n = 3). Therefore, we proceeded to examine the effects of pH on fertilization. Under acidic conditions (pH < 6.5), similar to those in the oviduct and spermiduct, fertilization was not observed within 3 h after insemination in either of the ascidians (Figure 1A,B). The fertilization rate increased as the pH increased above 7, and almost all eggs were fertilized at pH 8.2, which is the pH of seawater (Figure 1A,B). Thus, fertilization is inhibited in low pH conditions, such as those in the gonoducts, and an increase in the extracellular pH by contact with seawater is necessary for successful fertilization.

Ascidian spermatozoa are not highly motile when suspended in seawater and require activation by the sperm activating and attracting factors (SAAF) secreted by the eggs [24]. In seawater and low-pH conditions without eggs, the number of motile sperm is low, especially in *Phallusia* [25]. At pH 7.2, only 1.2% of the eggs were fertilized upon insemination with intact *Phallusia* sperm. However, the fertilization rate significantly increased when *Phallusia* sperm were pre-activated by exposure to pH 9.5 artificial seawater (ASW) before insemination (Figure 1B). Below pH 7.0, there was no improvement in the fertilization rate even with pre-activation of the sperm (Figure 1B). In *Ciona*, spermatozoa suspended in pH 6.5 ASW were also almost quiescent, but about 35% of the sperm suspended in pH 8.2 ASW showed progressive motility, although their velocity was slower than that of the activated sperm (Figure 1C). Interestingly, the velocity of the pre-activated *Ciona* sperm at all tested pH values was almost the same as in seawater (Figure 1C,D). A detailed analysis of the motility of pre-activated *Ciona* sperm at pH 6.4 and pH 8.2 using CASA showed hardly any significant differences in each parameter, except in beat-cross frequency (BCF) and path straightness (STR), which are relevant for hyperactivated sperm and/or sperm that have lost their straightness (Appendix A).

In the low pH conditions, where fertilization was not observed even with pre-activated sperm, few spermatozoa penetrated the vitelline coat (Figure 1E–G): the number of sperm penetrating the perivitelline space of the egg inseminated at pH 8.2 (control conditions) was 0.76 sperm/section, whereas only 0.002 sperm/section were found in the sections of the egg inseminated at pH 6.5 (Figure 1G). Since the fixed eggs had an average diameter of 110 µm, which corresponds to 18.3 sections, the estimated number of sperm that penetrated the vitelline coat under the control condition (pH 8.2) was 13.9 sperm/egg, and only 0.04 sperm/egg at pH 6.5, even after 3 h of insemination. Therefore, infertility under low pH conditions seems to occur partly because the spermatozoa are unable to penetrate the vitelline coat.

### 2.3. Incubation of Eggs at pH 8.2 before Insemination Induces Their Fertility

Similar to the above results (Figure 1), when the mature eggs of the ascidians were kept at lower pH conditions for 4 h and then inseminated with pre-activated sperm at the same pH, few eggs were fertilized. The fertilization rate at pH 7.0 was approximately 25%, and below pH 6.5 was almost 0% (Figure 2A,B). In contrast, when the eggs were incubated at pH 8.2 for 4 h and then inseminated at each of the experimental pH conditions, fertilization rates increased significantly; especially at pH 7, where the fertilization rate increased to 97% in *Ciona* and to 90% in *Phallusia* (Figure 2A,B).

The increase in fertilization rate by pre-incubation of the eggs at pH 8.2 was also observed in eggs in which the vitelline coat and accessory cells were removed (dechorionated eggs). The dechorionated eggs also showed low fertilization rates when inseminated with pre-activated sperm under lower pH conditions (Figure 2C). However, when the dechorionated *Phallusia* eggs were incubated in normal seawater at pH 8.2 for 4–6 h and then rinsed once with ASW with varying pH prior to insemination with the pre-activated sperm, the fertilization rate significantly increased at pH 6.0, 6.5, and 7.0 (Figure 2C).

This demonstrates that eggs remain infertile at lower pH values, and it suggests that contact with seawater upon spawning triggers egg fertility.

### 2.4. Effect of Ambient pH on pH_i_ of the Eggs

We examined the effect of changes in the ambient pH on the pH_i_ of the eggs by image analysis of BCECF-injected eggs. When the ambient pH around the eggs was increased from pH 5.7, which is the same pH as that in the oviducts, to pH 8.2, which is the same as that of seawater, the pH_i_ of the eggs gradually increased and reached a maximum at approximately 25 min exposure to the increased pH (Figure 2D). Furthermore, we found that once the pH_i_ of the eggs increased, the pH_i_ of the eggs did not decrease when the ambient pH returned to pH 5.7 (Figure 2D). We also determined the exact pH value of the eggs using ammonium acetate, which cancels the pH difference between the inside and outside of the cell. The pH of the eggs kept at pH 5.7 and those transferred to pH 8.2 was estimated to be pH 6.3–6.5 and pH 7.0–7.3, respectively (Appendix A). These results suggest that the spawning of eggs in seawater induces an increase in pH_i_ of the eggs, resulting in increased egg fertility.

### 2.5. Eggs under Low pH Conditions Do Not Release Sperm Attractants

Ascidian eggs release sperm attractants after oocyte maturation, and sperm chemotaxis has been observed [24,26]. To examine whether the eggs release sperm attractants under a low-pH condition, we examined the sperm-attracting ability of the egg-conditioned seawater prepared under varying pH conditions. In both *Ciona* and *Phallusia*, the egg-conditioned seawater prepared at pH 7 significantly reduced the sperm attracting activity, compared to that prepared at pH 8.2, and that at pH 6.5 further decreased sperm chemotaxis (Figure 3A,B). As mentioned above, pre-activated sperm can maintain motility even under low-pH conditions (Figure 1C,D). Similarly, low pH conditions did not affect the chemotactic behavior of sperm induced by the sperm activating and attracting factor (SAAF). *Ciona* spermatozoa showed obvious chemotactic behavior towards SAAF, even at pH 6.4. The pre-activated sperm did not show chemotaxis towards SAAF at pH 6.0, but this pH is lower than that of seminal plasma and body fluids and therefore outside of physiological conditions (Figure 3C). Furthermore, when the egg was punctured by a needle to artificially release sperm attractants, *Phallusia* spermatozoa showed potent chemotactic behavior towards it even at pH 6.5 (Figure 3D). These results indicate that the sperm attractant is not released under low-pH conditions, despite its presence in the eggs. This suggests that spawning results in an increase in the pH_i_ of the eggs, which triggers the release of the sperm attractant.

## 3. Discussion

This study demonstrated that the mature gametes of the ascidian species *Ciona intestinalis* and *Phallusia philippinensis* cannot be fertilized under the low-pH conditions found in the gonoducts. This suggests that the acidic conditions in the gonoductal fluids keep the eggs infertile and that the release of eggs into seawater increases the ambient pH, which enables egg fertilization. An increase in ambient pH triggers the release of sperm attractants from the egg, which is necessary for sperm chemotaxis. While the increase in the ambient pH has little effect on sperm fertility, the pH increase appears to be important for the initiation of sperm motility.

In general, the pH of body fluids in chordate is between 7.2 and 7.4, while the pH of seawater is approximately 8.2. Therefore, the pH_i_ of the eggs of marine animals showing external fertilization increases upon fertilization [27]. In this study, we found that the pH of the ascidian body fluids, especially in the oviduct, was much lower than that of other animals. This may be due to the effects of one type of ascidian blood cells called vanadocytes, which store sulfates in their vacuoles; the pH in the vanadocyte vacuoles ranges from 1.8 to 4.2 [28].

The effects of pH on fertilization have been studied in many species. In humans, the optimum pH for fertilization is 7.5 [29]. In the Japanese rice fish *Oryzias latipes*, fertilization is possible under a broad range of pH conditions (7.0–9.0) [30]. The fertilization rate is affected by the ambient pH, which also greatly affects the pH_i_ of gametes. In general, cells can regulate their pH_i_. In mammalian embryos, the HCO_3_^−^/Cl^−^ exchanger resists alkalinization, and the Na^+^/H^+^ exchanger (NHE) is involved in mitigating acidification [31]. However, these exchangers do not function in nascent oocytes, and oocytes do not regulate pH_i_ effectively. Therefore, their pH_i_ is affected by the ambient pH around the oocyte [22,31,32]. In sea urchins, the NHE is quiescent in unfertilized eggs; however, it becomes active within minutes of fertilization, resulting in an increase in pH_i_ and subsequent embryonic development [27]. NHE is also involved in the regulation of oocyte pH_i_ and maturation in starfish [22,33]. In this study, fertility and sperm attraction ability of ascidian eggs were significantly altered by changing the ambient pH. Present results show that the pH_i_ of the egg also increases with the increasing ambient pH, but does not change back when the ambient pH is lowered again. This suggests that the ability of ascidian eggs to regulate pH_i_ is incomplete, as observed in other animals, and that the egg pH_i_ depends on the ambient pH, with some regulatory ability retained by the egg. It is interesting to note that changes in ambient pH are essential to prepare eggs for fertilization.

The optimum pH of the trypsin-like enzyme, which mediates the penetration of the vitelline coat in the acrosomal vesicle of the ascidian *Halocynthia roretzi,* is 8.4 [34]. In our study, no sperm was found in the perivitelline space when *P. philippinensis* was inseminated at pH 6.5, indicating that a low pH may prevent the passage of sperm through the vitelline coat by inhibiting acrosomal enzymes. However, eggs that had their vitelline coat removed were also difficult to fertilize at pH 6.8 or below. This suggests that low-pH conditions may also prevent gamete membrane fusion. Incubation of unfertilized eggs at pH 8.2 prior to insemination significantly increased the fertilization rate, even if insemination was performed at pH 6.5–7.0. Furthermore, in vitelline coat-removed eggs, the fertilization rate at pH 6.0–6.5 was greatly improved by pre-incubation at pH 8.2. This suggests that fertility of the eggs is also mediated by the ambient pH. In starfish oocytes, an increase in intracellular pH caused by spawning unlocks the first meiotic metaphase I arrest and induces the progression of meiosis [20,22,33]. However, in ascidian eggs stored in the oviduct, meiosis is arrested at meiotic metaphase I, and this stage is preserved after spawning until fertilization [35,36]. Therefore, our results suggest that other oocyte maturation processes that are important for fertility may be affected by the ambient pH.

In this study, the chemotactic behavior of ascidian sperm toward the egg was not observed at pH values less than 7.5, as found in the gonoducts. Ascidian eggs release the sperm attractant SAAF [37,38], but mature ascidian oocytes in the ovary do not have sperm attracting ability despite the presence of SAAF in the oocyte [26]. Sperm chemotaxis was suppressed at a low pH because the secretion of SAAF from the egg is regulated by pH. Sperm attraction was observed even at a low pH when the eggs were punctured to mechanically release SAAF. This suggests that the sperm attractant factor SAAF was preserved in the eggs and that the release of SAAF from the eggs was activated by the pH increase.

Low ambient pH is known to inhibit sperm motility and fertility in some animals. The motility and fertility of human sperm decreases in media below pH 6.2 [9], and the sperm of sea urchins inside the male body are immotile due to the low pH environment [10]. In sperm, NHE, HCO_3_^−^/Cl^−^ exchangers, and carbonic anhydrase are involved in pH regulation, and gating of the sperm-specific channels CatSper and Slo3 is mediated by pH_i_ [3,15,17]. However, the velocity of motile ascidian sperm under low pH conditions was not significantly different from that under normal conditions, and chemotactic behavior of the sperm toward the sperm attractant SAAF was observed even at pH 6.4–6.5. Furthermore, when the eggs were pretreated at pH 8.2, fertilization was possible, even at pH 7.0. Hence, low ambient pH had relatively small effects on sperm motility and fertility in ascidians, and the decrease in fertilization rate caused by a low ambient pH appeared to be mainly due to mechanisms involved within the egg. However, acidic conditions, such as pH 6, also inhibited sperm motility, and high-pH treatment initiated sperm motility. Therefore, an increase in ambient pH at the time of spawning plays a role in the initiation of sperm motility. However, this study did not confirm whether the increase in pH_i_ of the oocyte really enhances release of SAAF. Furthermore, there is a significant difference in the parameter of beat-cross frequency (BCF) in *Ciona* sperm. Thus, some hyperactivation-like phenomenon may be mediated by pH-dependent CatSper, as in mammals. As there is no knowledge of ascidian sperm capacitation and hyperactivation, we would like to examine this particular aspect in future studies.

## 4. Materials and Methods

### 4.1. Materials

Specimens of the ascidian species *Phallusia philippinensis* (misidentified as *P. nigra* in previous papers [39]) were collected from the Ginowan Fishery Bay on the west coast of Okinawa Island, Japan. *Ciona intestinalis* (type A: also called as *C. robusta*) were supplied by National Bio-Resource Project at Misaki Marine Biological Station, University of Tokyo. *P. philippinensis* specimens were kept in an aquarium at 20–25 °C in the dark, whereas *C. intestinalis* specimens were kept at 16 °C under constant light until experimental use. Eggs and semen were collected from the oviduct and spermiduct, respectively, via dissection. Semen was stored on ice or at 4 °C until further use. Eggs were placed in either artificial seawater (ASW) (pH 8.2) or measuring medium after collection and immediately used for experiments. Where dechorionated eggs were required, the vitelline coat and accessory cells were manually removed from the eggs using a sharpened insect pin and fine blade (Shiga Konchu, Tokyo, Japan). Dechorionated eggs were kept in a 1.5% agar-coated dish to avoid disruption.

The ASW consisted of NaCl (462 mM), KCl (9.2 mM), CaCl_2_ (9 mM), MgSO_4_ (28 mM), MgCl_2_ (22 mM), and 10 mM HEPES (Dojindo, Kumamoto, Japan) (pH 8.2). The measuring media were: ASW, whose pH was adjusted with 10 mM good buffers (Dojindo): MES (pH 5.0–6.5); pH PIPES (pH 6.5–7.2); HEPES (pH 7.2–8.2); TAPS (pH 8.2); CHES (pH 9.5) instead of HEPES (pH 8.2).

This study using invertebrates is not regulated by animal welfare, but experiments were performed in accordance with the principle of animal welfare.

### 4.2. Pre-Activation of Sperm Motility

The density of sperm in the undiluted semen was calculated by counting the number of diluted sperm with a hemocytometer. Since *Phallusia* sperm are almost quiescent even if suspended in ASW (pH 8.2) [25], sperm motility was pre-activated by suspension of semen into 100× the volume of the medium (pH 9.5 ASW) for 10–15 min. Then, the pH was re-adjusted to experimental conditions by diluting the activated sperm more than 100-fold in the measuring medium before insemination. Pre-activation of *Ciona* sperm was induced by 1 mM theophylline, as described elsewhere [40].

### 4.3. Evaluation of Fertilization

Eggs were inseminated with 1 × 10^7^ cells/mL sperm suspension in the measuring medium. The fertilization rate was evaluated by counting the number of embryos that had developed to the 2-cell stage or a later stage 3 h after insemination. Over 300 eggs were assayed to determine the fertilization ratio.

### 4.4. Measurements of pH

Oviductal fluid and semen were collected from mature adult ascidian bodies by dissection, and their pH values were measured using a pH meter (HM-40V, TOA-DKK, Tokyo, Japan) equipped with a fine electrode (GT5426-S, TOA-DKK). Blood was collected from the heart, and its pH was measured using a pH meter (Twin pH B212, HORIBA, Kyoto, Japan). Intravital organ pH was measured using a pH datalogger (SensorLink PCM700, ThermoFisher, Waltham, MA, USA) with a stainless-steel pH electrode (Orion 9863BN, ThermoFisher).

Measurements of pH changes in eggs were performed by imaging with the pH-sensitive dyes 3′-O-Acetyl-2′,7′-bis(carboxyethyl)-4 and 5-carboxyfluorescein (BCECF). BCECF-dextran (Mr = 10,000) (ThermoFisher) was dissolved at 500 mg/mL in 100 mM K aspartate and microinjected into eggs to obtain a cytoplasmic concentration of 5–15 mg/mL. Eggs were exposed to 440 nm and 490 nm excitation lights, and fluorescence images at 520 nm were taken using an inverted microscope (Diaphot TMD300, Nikon, Tokyo, Japan). Images were captured using a digital camera BS41L (BITRAN, Saitama, Japan) and the ImageJ software (Ver.1.48-1.52; NIH, Bethesda, MD, USA). Changes in the intracellular pH were evaluated using the ratio of the fluorescence intensities at 440 nm (F_440_) and 490 nm (F_490_). At the end of each experiment, the pH_i_ of the eggs was lowered by replacing the medium with 20 mM ammonium acetate (in 10 mM MES, pH 5.7) to confirm that the pH_i_ change was measured by imaging.

### 4.5. Analysis of Sperm Motility and Chemotaxis

To examine the release of the sperm attractant from the eggs, the egg-conditioned seawater was used [25]. One volume of eggs was suspended in 15 volumes of ASW with different pH and incubated for 16–20 h at 4 °C. The egg suspensions were centrifuged at 20,000× *g* for 15 min at 4 °C and the obtained supernatant was designated as the egg-conditioned seawater.

Sperm motility and chemotaxis were examined, as previously described [40]. Pre-activation of sperm motility was performed, as described above. The pre-activated sperm suspension was mounted on the observation chamber pretreated with bovine serum albumin, and sperm movements around a micropipette tip containing the egg-conditioned seawater or 1 µM SAAF were recorded with a digital camera (Nikon1, Nikon, Tokyo, Japan) attached to a BX51 microscope (Olympus, Tokyo, Japan) under darkfield illumination. To analyze sperm chemotaxis around the egg, the dechorionated egg was placed directly in the observation chamber. For the chemotaxis assay, forced punctuation of the eggs was performed using a glass needle. The velocities and trajectories of spermatozoa were analyzed using the Bohboh software (BohbohSoft, Tokyo, Japan) [41]. The linear equation-based chemotaxis index (LECI) was calculated, as described previously [42].

The motility parameters, including curvilinear velocity (VCL) of the *Ciona* sperm, were observed by a microscope (BX 51, Olympus) equipped with the CCD camera (acA1300-200uc, Basler, Ahrensburg, Germany), and analyzed using the CASA system (SCA, Microptic, Barcelona, Spain) with sperm counting chamber slides (SC20-01-04-B, Leja, Nieuw-Vennep, The Netherlands). Spermatozoa with STR > 80 (%) and VCL > 10 (µm/s) were classed as motile. For precise information on the individual movement parameters, please see [43].

### 4.6. Evaluation of the Sperm Entry into the Perivitelline Space

The eggs were inseminated with pH 9.5 ASW-activated sperm for 15 min and fixed in 10% formalin in 50% ASW (pH 8.2). The fixed samples were dehydrated using an alcohol series, embedded in paraffin, and cut into 6 µm thick sections. The sections were stained with 0.1 µg/mL 4′,6-diamidino-2-phenylindole (DAPI) and 0.05 mg/mL fluorescein isethionate (FITC) in 50% ethanol to detect the nucleus and vitelline coat, respectively. The stained sections were mounted with Gel Mount (Cosmo Bio, Tokyo, Japan) and observed using a fluorescence microscope (BX 51 Olympus) equipped with a filter block U-MWU2 (Olympus) so that both the sperm heads and the vitelline coat could be seen simultaneously. The number of sperm in the perivitelline spaces was counted for each section, and the number of sperm that penetrated the vitelline coat per egg was calculated. Nuclei that appeared as rods when stained with DAPI were counted as spermatozoa.

### 4.7. Statistical Analysis

All experiments were repeated at least three times with different specimens. Data are expressed as the mean ± standard deviation. Statistical significance was calculated using the Student’s *t*-test or Dunnett’s test; *p* < 0.05 was considered significant.

## 5. Conclusions

In this study, we found that acidic conditions in the body fluids of the Phlebobranchiata ascidians, *C. intestinalis* and *P. philippinensis*, prevent the fertilization of mature gametes in the gonoducts. The study revealed that after spawning the increase in ambient pH caused by contact with seawater increases the pH_i_ of the eggs, which induces fertility and sperm attraction of the eggs. This is the first study to show that pH_i_ changes at spawning play an important role specifically in egg fertility, even though many studies have shown the role of pH changes in fertilization. In addition, we have demonstrated that pH also affects motility of the sperm. However, there has been no clear effect of the pH increase during spawning. Further studies are necessary to understand the role and molecular mechanism of pH changes in gamete fertility.

## Figures and Tables

**Figure 1 ijms-24-02666-f001:**
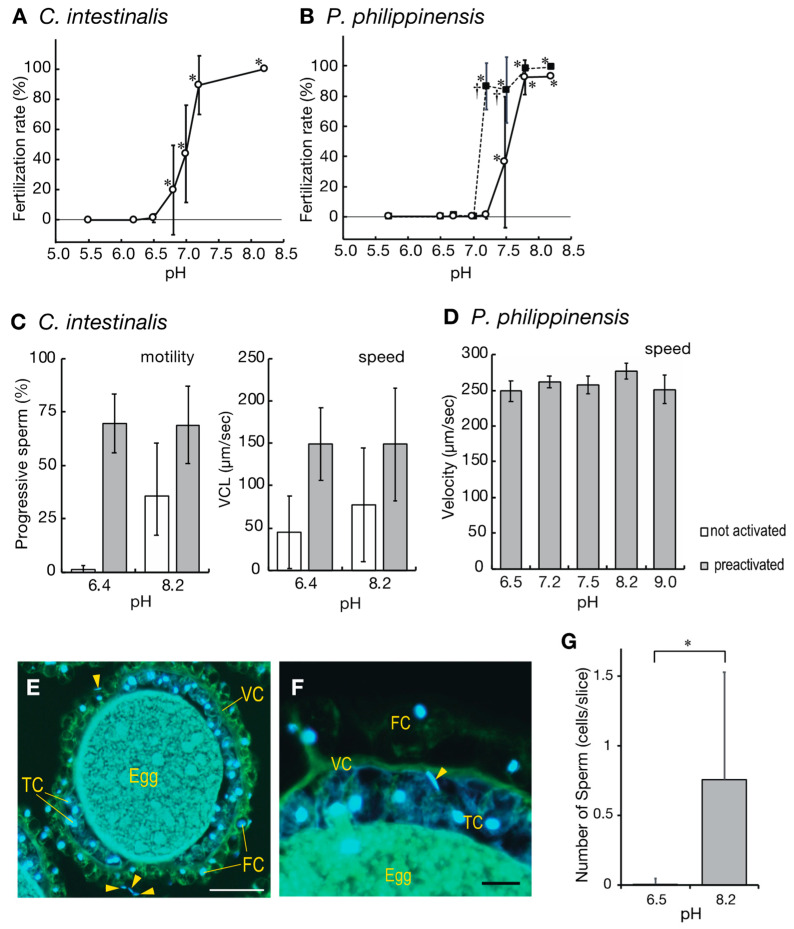
(**A**,**B**) Effects of pH on the fertilization rates of the ascidians *Ciona intestinalis* (**A**) and *Phallusia philippinensis* (**B**). Fertilization rates of eggs inseminated with intact sperm (open circle) or pre-activated sperm (closed square) at 3 h after insemination. Values are expressed as the mean ± standard deviation (SD) of four experiments. Asterisks indicate the point of statistical significance against the fertilization rate at pH 6.5 (*p* < 0.05; Dunnett’s test). Daggers indicate the point that has statistical significance against the fertilization rate between activated and non-activated sperm (*p* < 0.05; Student’s *t*-test). (**C**,**D**) Effects of pH on sperm motility in *C. intestinalis* (**C**) and *P. philippinensis* (**D**). White bars show non-activated sperm, gray bars show sperm pre-activated by 1 mM theophylline (*Ciona*) or pH 9.5 artificial seawater (*Phallusia*). VCL: curvilinear velocity. Speed was calculated from the progressively motile sperm. Values are expressed as the mean ± SD of four experiments. (**E**–**G**) Evaluation of sperm entry into the perivitelline space of the inseminated eggs of *P. philippinensis.* Eggs fixated 3 h after insemination were cut into 6-µm-thick sections to observe the sperm inside the perivitelline space. Sperm nuclei stained with DAPI are rod-shaped (arrowhead) (**E**,**F**). Scale bars indicate 100 µm (**E**) or 10 µm (**F**) thickness. VC: vitelline coat; TC: test cells; FC: follicular cells. (**G**) Number of sections showing sperm nuclei in the perivitelline spaces of the inseminated egg at pH 6.5 and 8.2. Since a fixated egg consisted of an average of 18.3 sections (diameter of the egg: 110 µm), the number of sperm penetrating the vitelline coat was estimated to be 13.9 sperm/egg at pH 8.2 and 0.04 sperm/egg at pH 6.5. Asterisks indicate statistical significance (*p* < 0.05; Student’s *t*-test).

**Figure 2 ijms-24-02666-f002:**
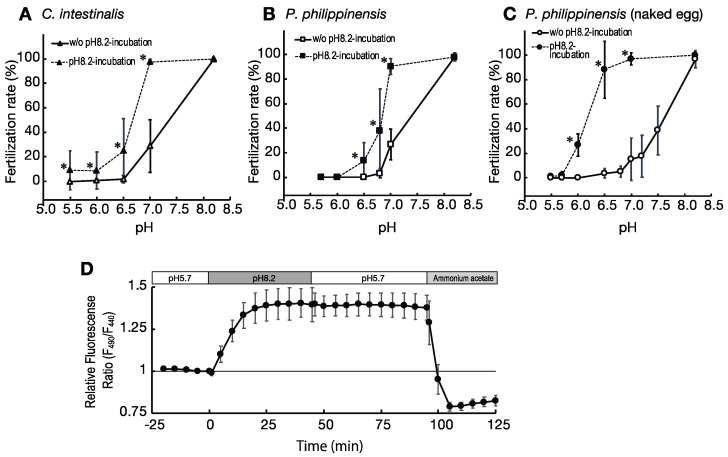
Effect of the pH on fertilization rates of eggs pre-incubated at pH 8.2 before insemination, in ascidians *Ciona intestinalis* (**A**) and *Phallusia philippinensis* (**B**,**C**). Closed marks show fertilization rates of eggs pre-incubated at pH 8.2 for 4 h. Open marks show fertilization rates of eggs pre-incubated in the insemination medium for 4 h. Values are expressed as the mean ± standard deviation (SD) of three to six experiments. Asterisks indicate the point that has statistical significance between the eggs with or without pH 8.2-incubation (*p* < 0.05; Student’s *t*-test). (**D**) Intracellular pH changes in an egg were analyzed using the fluorescent pH sensor BCECF. Values are expressed as the mean ± SD of 15 eggs from three specimens. The ambient pH is indicated by the box in the upper part of the graph.

**Figure 3 ijms-24-02666-f003:**
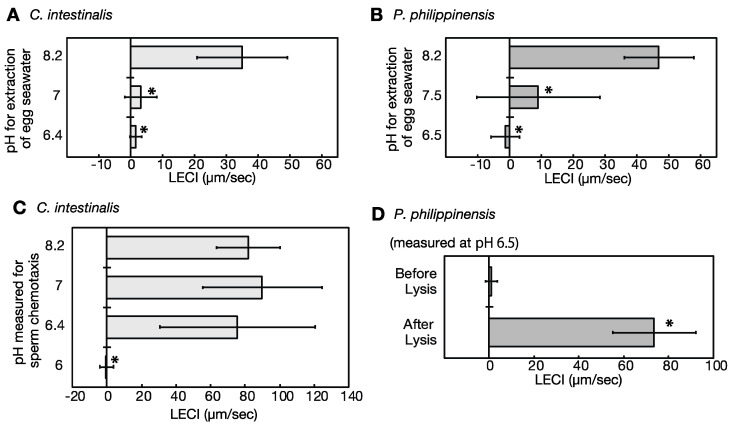
Effect of pH on sperm chemotaxis of the ascidians *Ciona intestinalis* and *Phallusia philippinensis*. (**A**,**B**) Effect of pH on sperm attracting ability of the eggs of *C. intestinalis* (**A**) and *P. philippinensis* (**B**). The egg-conditioned seawaters were prepared at a different pH, packed in capillary tips, and sperm movements in the vicinity were observed to calculate the chemotaxis index (LECI). Values are expressed as the mean ± standard deviation (SD) of three samples (30–36 sperm). Asterisks show the point of statistical significance against LECI at pH 8.2 (*p* < 0.05, Dunnett’s test). (**C**) Effect of pH on chemotactic behavior of the sperm of *C. intestinalis*. The spermatozoa were pre-activated with 1 mM theophylline, and sperm movements around the capillary containing SAAF, the sperm attractant of *C. intestinalis,* were examined in each pH conditions. Values are expressed as the mean ± SD of ten samples (10–11 sperm). Asterisks show the point of statistical significance against LECI at pH 8.2 (*p* < 0.05, Dunnett’s test). (**D**) Sperm attracting ability of intact and punctured eggs of the ascidian *P. philippinensis*. Values are expressed as the mean ± SD of four samples (31–41 sperm). Statistical significance is set at * *p* < 0.05 (Student’s *t*-test).

**Table 1 ijms-24-02666-t001:** Ascidian tissues and fluids: pH.

Tissue/Fluid	*Ciona intestinalis*	*Phallusia philippinensis*
Tissue		
oviduct	6.36 ± 0.16 (12)	5.65 ± 0.18 (7)
spermiduct	6.66 ± 0.08 (7)	6.44 ± 0.14 (4)
ovary	6.71 ± 0.07 (9)	No data
heart	6.82 ± 0.09 (10)	No data
Fluid		
blood	6.49 ± 0.12 (31)	6.20 ± 0.19 (14)
oviductal fluid	6.18 ± 0.18 (32)	5.69 ± 0.14 (18)
semen	6.53 ± 0.13 (9)	6.65 ± 0.18 (17)

Data are presented as the mean ± standard deviation (SD). The numbers of samples are shown in parentheses.

## Data Availability

All data presented in this study are available in the article.

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
