# Peer review of "Spawning-Induced pH Increase Activates Sperm Attraction and Fertilization Abilities in Eggs of the Ascidian, Phallusia philippinensis and Ciona intestinalis"

_ijms, 2023, doi:10.3390/ijms24032666_

Round 1

Reviewer 1 Report (Previous Reviewer 2)

This article is very oryginal and describes effects of pH on sperm function and egg fertilization in ascidians. The description is clear however I sugest some modificatiins

1.add information on studied species into the title

2.provide number of animals usied for the study,were they in similar age?
3. Describe biology in more detailed way of studied species

4.add information on study limitations

5.add permit number for this study

6.it will be nice to have bigger microphotographs

7.give more detailed information on studied parametrem of motility , one is mentioned but not described

Author Response

This article is very oryginal and describes effects of pH on sperm function and egg fertilization in ascidians. The description is clear however I sugest some modificatiins

(answer)
Thank you very much for the positive comments. We have corrected the points you pointed out as follows.

1.add information on studied species into the title

(answer)
We have added the scientific names of the used species into the title.

2.provide number of animals usied for the study,were they in similar age?

(answer)
The number of individuals used is already listed in the form of "number of experiments". It is difficult to define "similar age" of ascidians because their lifespan is about 4 to 6 months, but we used specimens that were generally 2 to 3 months after settlement.

3.Describe biology in more detailed way of studied species

(answer)
We have added the sentences in Introduction (lines 87 - 90) as follows:

“Ascidians are marine invertebrates belonging to the primitive chordates. Especially Ciona intestinalis, due to its a small genome, its whole genome was elucidated in 2002 [21], and now it has been one of the most used marine invertebrates as a model organism for fertilization and developmental studies in recent years.”

4.add information on study limitations

(answer)
We have added the sentence about limitation of the study in Discussion (Lines 366-367), in accordance with the reviewer’s suggestion: “However, this study did not confirm whether the increase in pHi of the oocyte really enhances release of SAAF.”

5.add permit number for this study

(answer)
There is no permission number, because this research is on invertebrate animals and is not under animal welfare regulations. However, the research is being performed in accordance with the spirit of animal welfare for vertebrate animals. This has been added to Materials and Methods. (Lines 394-395)

6.it will be nice to have bigger microphotographs

(answer)
We have enlarged the photographs in Fig. 1.

7.give more detailed information on studied parametrem of motility, one is mentioned but not described

(answer)
To explain the details and meaning of the individual CASA parameters would be a long text and difficult to explain without figure. Therefore, we have added a citation of the review on CASA to Materials and Methods, and Supplements. Only BCF and STR, which are reffered in the results, we added a brief explanation (Lines 151-152).

Reviewer 2 Report (New Reviewer)

In my opinion, the authors are right and pH plays a very important role in the process of gamete activation and fertilization itself. Since there are not many publications in this field, this publication is very important and I hope that it will draw the attention of other researchers to this problem. Despite the corrections, this manuscript still needs some corrections. My detailed comments/suggestions are included in the MS text. To see them all, open the file in Acrobat Reader. I believe the authors will have no problems improving this MS. After taking into account the corrections, I will recommend accepting this article for publication.

Author Response

In my opinion, the authors are right and pH plays a very important role in the process of gamete activation and fertilization itself. Since there are not many publications in this field, this publication is very important and I hope that it will draw the attention of other researchers to this problem. Despite the corrections, this manuscript still needs some corrections. My detailed comments/suggestions are included in the MS text. To see them all, open the file in Acrobat Reader. I believe the authors will have no problems improving this MS. After taking into account the corrections, I will recommend accepting this article for publication.

(answer)
Thank you very much for the positive comments. We have corrected the points suggested by the reviewer. Precise corrected points and answers to the reviewer’s suggestion were shown in the attached pdf file.

Round 2

Reviewer 2 Report (New Reviewer)

I recommend accept the MS at present form

This manuscript is a resubmission of an earlier submission. The following is a list of the peer review reports and author responses from that submission.

Round 1

Reviewer 1 Report

Comments to Authors:

OVERVIEW AND GENERAL RECOMMENDATION:

In the present study the authors investigated the effect of pH on fertilization of two ascidian species (Phallusia philippinensis and Ciona intestinalis), which have external fertilization. The authors hypothesized that the basic pH of seawater (in relation to body fluid) triggers gamete activation and enables fertilization. First, they observed a higher pH in seawater (8.2) compared to the fluid in the gonoducts (5.5-6.8). Then they investigated the effect of seawater pH on fertilization rates, sperm motility and sperm chemotaxis. They concluded that the basic pH of seawater enables fertilization by regulating the oocyte sperm-attracting activity, but has little effect on sperm motility and fertilizing ability.

This study increases the current knowledge of fertilization in ascidians, in particular the importance of fluid pH for the fertility of gametes. However, the conclusions of this study are not fully supported by the data shown. Moreover, the methods are not adequately described and miss some relevant information. Last, I believe the findings are superficial and the authors should have expanded their experiments to reach a better understanding of how pH and other fluid characteristics affect fertilization. Therefore, I cannot recommend this paper for its publication in its present form. Below there is a detailed explanation of my major and minor concerns about this paper.

MAYOR CONCERNS

-          The conclusions of this paper are not fully supported by the results. The authors conclude that the major effect of pH on fertilization is on oocyte fertility, in specific its ability to release sperm-chemoattractants, and not on the sperm fertility and motility. However:

o   The fertilization rate at pH 7 is improved when sperm is preincubated at higher pH (Figure 1), which indicates an effect of pH on sperm acquisition of fertilizing ability.

o   Less number of sperm were active in the low pH seawater, which indicates an effect on motility (explained in part 2.2, although not in the figure).

o   Sperm chemotaxis towards SAAF is affected at pH 6 (Figure 3).

o   Sperm penetration of the perivitelline coat is also affected and with the experiment performed it is not possible to elucidate if the problem is on the sperm or on the oocyte (Figure 1).

o   Even if motility was not affected that does not mean it could not affect other sperm parameters important for fertilization. This should be further discussed.

o   It should be acknowledged that pH may be affecting fertility by interfering with different maturation mechanisms both of the sperm and the oocyte. With the experiments performed it is not possible to discard the effect on sperm or identify other effects on the egg besides the release of chemoattractants.

-          I believe some additional experiments would improve the quality of the study:

o   The authors should test if pre-incubation of sperm and or oocytes with sea-water at normal pH improves the sperm penetration into the perivitelline space. This would allow us to know if the pH is affecting the ability of sperm to penetrate or the ability of the egg to be penetrated.

o   The authors should expand the analysis of sperm motility beyond velocity. Are there other motility parameters that are affected by pH that could be important for fertilization?

o   It would have been interesting to include other experiments that further study mechanisms of action of pH on sperm and egg fertility. For instance, if CatSper activation is involved or if cytoplasmic maturation is affected.

-          There is no explanation of the statistics used to analyze the data. Please, add a section in the methods to explain. Further, in figures 1 and 2 there is no p-value or indication of statistical differences between pH conditions and pre-activation. Please, include those statistics on the results.

-          The methods lack important information to understand the experiment and experimental design. Please explain better the treatments used and experiments performed. Also explain better the following:

o   4.2. Expand the methods about how the fertilization was performed (media, culture time, time when 2-cell embryos were evaluated, etc).

o   4.3. Is it possible to use some standards to extrapolate the exact pHi of eggs, instead of just observing an increase or decrease? How many eggs were assessed per experiment?

o   4.4. Expand these methods and specify the different experiments performed. Was SAAF used for assessing motility in the first experiment or just for chemotaxis? What kind of observation chamber was used? How many sperm were evaluated per treatment? How was LECI calculated? Also, explain the methods for the experiment about chemotaxis towards eggs (whole and punctured)?

o   4.5. How was the number of sperm in the perivitelline space per egg calculated? How many eggs were evaluated?

-          The introduction fails to give enough background to understand the present study. Specifically it does not state clearly what is the lack of knowledge that the study wants to fill, what is the aim and why it is important to study it. Further, I would recommend starting the introduction with a brief explanation of general characteristics of the species and their reproduction, as there may be some readers interested in fertilization that are not familiar with the species.

-          The discussion is hard to read through. I would recommend organizing it better for discussing the main findings. Moreover, it should be further discussed how pH can affect fertility beyond the main finding of the study (as other effects could be happening at the same time). Last, it would be relevant to mention how other characteristics of seawater beyond pH (for instance osmolality) could be important for triggering fertility.

MINOR CONCERNS

-          Abstract: Please, add aims and conclusions in the abstract. Also, make it clear when referring to past and present study.

-          L. 17: Change Fertilized for Fertilize

-          L. 18-21: Is this in previous studies? Please specify.

-          L. 21: Before <<The pH values>> add <<In the present study,>>.

-          L. 23: Change remained for was.

-          L. 27: Change triggers for enables. The pH itself does not fertilize the egg, but allows for fertilization to take place).

-          L. 27: Before <<An increase in>> add <<We also found that>>.

-          L. 32-36: Instead of this, start with something more specific to the species studied.

-          L. 42: Change in oviduct  to in the oviduct.

-          L. 58-60: Rephrase to <<The K+ channel Slo3 is also a sperm-specific channel that is regulated by pHi and that may regulate sperm function through changes in pH and membrane potential>>.

-          L. 69-70: Erase <<because the molecular mechanisms involved in oocyte maturation and fertilization are complicated>>.

-          L 71: Change were for are.

-          L. 72: Change were for are.

-          L. 71-77: Add references.

-          L. 79-81: The oocyte maturation process was not studied. Please change this sentence to something that refers to fertilization.

-          L. 83: Change this title to <<pH in different ascidian organs and fluids is acidic and lower than sea water>>. Keep low is not correct because it was not tested if the pH was maintained in different conditions and low is relative.

-          L. 88: Did the authors also study Ascidia sydneiensis? Where are those results?

-          L. 90-92: This sentence is repetitive of the previous lines. Please summarize better the results in Part 2.1.

-          L. 92-93: Change pH is maintained at an acidic level in not only for pH is acidic not only.

-          L. 94: Change remain for are kept.

-          Table 1: Change pH in the ascidian tissues for pH in ascidian tissues and fluids

-          Table 1: Remove the * in bot the the table and the footnote.

-          L. 99: Change Perivitelline space pH of Phallusia eggs for The pH of the perivitelline space in Phallusia eggs.

-          L.104-107: Add reference

-          L. 113: change by dilution to by contact.

-          L. 116: Define what SAAF is.

-          L. 116-117: Where is this data shown? In the corresponding figure, I do not see results of the number of sperm activated in the different pH. Please include this data. Also define what activated means. Do sperm need to show a specific type of motility or any kind of motility to be considered activated? Change were small for was low. Add a dot after Phallusia and start a new sentence.

-          L. 118-119: Change sentence for <<At pH 8.2 the number of activated spermatozoa was also low even though the activated spermatozoa showed almost the same velocity as that in seawater>>. 

-          L. 121: What did the pre-activation consist of? Cultured in which conditions and for how long?

-          L. 124: Add a comma after conditions.

-          L. 126: the perivitelline space.

-          L. 136: change the eggs inseminating for the number of eggs inseminated. Please specify how insemination was assessed (the number of 2-cell embryos?).

-          L. 143: Change entering into for inside. Erase <<(see Materials and Methods)>>.

-          L. 144-149: I believe the way this data is presented is confusing. I suggest explaining this in the methods and for the figure simply present the number of sperm in perivitelline space per oocyte. Also, include error bars in this figure.

-          L. 160: Erase the comma.

-          L. 163-164: A suppression of egg fertility is a strong statement. Better erase this sentence as there is already a conclusion of this result part in L. 168.

-          L. 165: Why was the preincubation time variable?

-          L.171: Define BCECF-.

-          L. 179: Erase <<resulting in fertilization>> which is not evaluated in this specific part and in any case the increase of pHi does not induce fertilization but egg fertility.

-          L. 183: Was the preincubation 2 h, as mentioned here, or 4 h, as mentioned in the text above?

-          L. 183-184: Open marks refer to non-preincubated eggs. Correct this sentence accordingly.

-          L. 193: Change <<and that at pH 6.5 had little activity>> for <<and that at pH 6.5 further decreased sperm chemotaxis>>.

-          L. 195: change furthermore for similarly.

-          L. 198: add <<to artificially release the sperm chemoattractants>>.

-          L.204: change in the eggs for of the eggs

-          L. 206: Erase the comma and <<(see Materials and Methods)>>.

-          L. 206-207: Change <<on chemotactic behavior of the sperm>> for <<on the chemotactic behavior of the sperm of C. intestinalis>>.

-          L. 207: Add <<were pre-activated with medium at different pH>>.

-          L.207-208: What is the sperm attractant of C. intestinalis and was this also used at the same time than SAAF or was it a separate experiment?

-          L. 208-209: Change <<Activity of the sperm attraction of intact and lysed egg>> to <<Sperm-attracting activity of intact and lysed eggs>>.

-          L. 234: Change Normally for In general.

-          L.243: This claim is inaccurate. Your results also indicate that once the pH is up it does not come back down when exposed to acidic medium. Therefore there is some kind of pH regulation occurring or the pHi of eggs does not completely depend on ambient pH.

-          L. 245: Specify the enzyme.

-          L. 246: Change Furthermore for In our study.

-          L.213-218: This first paragraph is confusing and inaccurate as low pH does not trigger egg fertilization but enables it. Moreover, the chemotaxis towards SAAF was affected at pH 6 and the number of activated sperm was also affected by pH, so sperm motility is affected by pH. I would suggest rephrasing to: In this study, it was revealed that the gonoductal gametes of ascidian species are maintained under low-pH which keeps the eggs infertile, and that spawning to seawater increases the ambient pH enabling fertilization. Moreover the study showed that the increase in ambient pH is necessary for sperm chemotaxis as it triggers the release of sperm attractants from the egg.

-          L. 220: Erase <<which is much higher than that in body>>.

-          L. 224: change the vacuoles for their vacuoles.

-          L. 225-226: This is confusing. If pH is acidic in all the body parts, not just the blood, better to erase this hypothesis of the pH being theoretically higher.

-          L.250: may also prevent.

-          L. 251: Change However for On the other hand

-          L. 257-259: Rephrase to <<However, in ascidian eggs stored in the oviduct, meiosis is arrested at meiotic metaphase I and this stage is preserved after spawning until fertilization>>.

-          L. 259-261: Rephrase to <<Therefore, our results suggest that other oocyte maturation processes important for fertility may be affected by ambient pH>>.

-          L.263: Change <<was not observed>> t<<is not present>>.

-          L. 266: change <<in male body>> to<<inside the male body>>.

-          L. 269-271: This statement is inaccurate because your results indicated differences in the number of active sperm, and lower chemotaxis at pH 6.

-          L.272: What does almost means? Either there was fertilization or not.

-          L. 273-275: Based on your results, this statement is not correct.

-          L. 279: Change <<when the eggs were disrupted>> to <<when the eggs were punctured to mechanically release SAAF>>.

-          L. 280-281: This is a hypothesis. The SAAF transporters were not investigated. Instead, mention that the pH affects SAAF release. And if the transporters are mentioned, give more information about how they are involved in SAAF release and include references.

-          L. 290: What temperature was the sperm kept at? What is measuring medium, is it the treatment medium? Please rephrase here and in the rest of the methods, and specify that treatments consist of different pH.

-          L. 291: Change <<In few experiments, vitelline coat>> to <<In some experiments, the vitelline coat>>

-          L. 292: Change glade for blade.

-          L. 291-293: Include that these eggs are called dechorionated as in the Results section.

-          L. 327: Change measurements for treatments.

-          L. 343. The number.

-          L. 344. the number.

-          L. 347. Change <<by spawning>> to <<by contact with seawater after spawning>>.

-          L. 349 -350: Rephrase this sentence.

-          L.351-353: In which species?

-          L. 354: Change study to understand.

Author Response

OVERVIEW AND GENERAL RECOMMENDATION:

In the present study the authors investigated the effect of pH on fertilization of two ascidian species (Phallusia philippinensis and Ciona intestinalis), which have external fertilization. The authors hypothesized that the basic pH of seawater (in relation to body fluid) triggers gamete activation and enables fertilization. First, they observed a higher pH in seawater (8.2) compared to the fluid in the gonoducts (5.5-6.8). Then they investigated the effect of seawater pH on fertilization rates, sperm motility and sperm chemotaxis. They concluded that the basic pH of seawater enables fertilization by regulating the oocyte sperm-attracting activity, but has little effect on sperm motility and fertilizing ability. 

This study increases the current knowledge of fertilization in ascidians, in particular the importance of fluid pH for the fertility of gametes. However, the conclusions of this study are not fully supported by the data shown. Moreover, the methods are not adequately described and miss some relevant information. Last, I believe the findings are superficial and the authors should have expanded their experiments to reach a better understanding of how pH and other fluid characteristics affect fertilization. Therefore, I cannot recommend this paper for its publication in its present form. Below there is a detailed explanation of my major and minor concerns about this paper.

(Response)
Thank you very much for your very useful and extensive remarks. Many of your points are very important and we are very appreciating them.

Many points were indicated on the conclusion of the study that increase in extracellular pH does not contribute to motility including chemotactic behavior and fertility of the spermatozoa.  It is true that the ascidian sperm, especially the Phallusia sperm, are greatly affected by pH. The Phallusia sperm are almost quiescent in seawater without a factor from the egg. Only spermatozoa treated at pH 9.5 initiate motility, and continue to move even though pH in the external fluid is changed.
This fact had been described in the Introduction and results in the original manuscript. However, we realized that it was not conveyed at all, probably because we emphasized the effects of pH on eggs in order to clarify message of the manuscript. Thus, we have revised the manuscript to reflect that.

Due to several requests from the reviewer for additional experiments, we have planned the experiments, but unfortunately, the period for obtaining Phallusia gametes has finished this year, and we will not be able to do the experiments until next year.

Since we cannot add more experiments even if we extended the revise period, please allow us to make the paper only what we can say with results in the present manuscript. We have added some experiments that we have already done and did not include in the manuscript. We are very sorry that we could not meet your expectations.

MAYOR CONCERNS

  • The conclusions of this paper are not fully supported by the results. The authors conclude that the major effect of pH on fertilization is on oocyte fertility, in specific its ability to release sperm- chemoattractants, and not on the sperm fertility and motility. However:
  • The fertilization rate at pH 7 is improved when sperm is preincubated at higher pH (Figure 1), which indicates an effect of pH on sperm acquisition of fertilizing ability.
  • Less number of sperm were active in the low pH seawater, which indicates an effect on motility (explained in part 2.2, although not in the figure).
  • Sperm chemotaxis towards SAAF is affected at pH 6 (Figure 3).
  • Sperm penetration of the perivitelline coat is also affected and with the experiment performed it is not possible to elucidate if the problem is on the sperm or on the oocyte (Figure 1).
  • Even if motility was not affected that does not mean it could not affect other sperm parameters important for fertilization. This should be further discussed.
  • It should be acknowledged that pH may be affecting fertility by interfering with different maturation mechanisms both of the sperm and the oocyte. With the experiments performed it is not possible to discard the effect on sperm or identify other effects on the egg besides the release of chemoattractants.

(Response)
As described above, the Phallusia sperm are almost quiescent in seawater without a factor from the egg, and only high-pH-treated spermatozoa continue to move. We have revised the entire document to clarify this point.

However, the main point of this manuscript is the implications of the pH change observed during spawning (pH 5.5 > 8.2) on fertilization. The pH change in this range does not lead to sperm activation, whereas if sperm are activated at high pH, sperm motility is continued even at low pH. Therefore, the conclusion of this manuscript remains that the main target of the change is the egg.

- I believe some additional experiments would improve the quality of the study:

  • The authors should test if pre-incubation of sperm and or oocytes with sea-water at normal pH improves the sperm penetration into the perivitelline space. This would allow us to know if the pH is affecting the ability of sperm to penetrate or the ability of the egg to be penetrated.

(Response) Because preincubation of sperm with seawater (pH 8.2) do not activate sperm motility without eggs, so we did not conduct the experiment.

  • The authors should expand the analysis of sperm motility beyond velocity. Are there other motility parameters that are affected by pH that could be important for fertilization?

(Response) Thank you very much for your useful suggestions. However, as mentioned above, we are unable to conduct additional experiments because the reproductive season for ascidians has finished this year. In addition, since the purpose of this study is not to investigate the role of pH on the fertility of sperm, we have concluded that we do not add these experiment at this time.

We would like to conduct the research as an extension of this study from next year.

  • It would have been interesting to include other experiments that further study mechanisms of action of pH on sperm and egg fertility. For instance, if CatSper activation is involved or if cytoplasmic maturation is affected. 

(Response) We completely agree the reviewer's comment, and we would like to develop this study on role of Catsper in the future. However, as stated above, we have no choice but to give up on this additional experiment. We apologize for not meeting your expectations.

  • There is no explanation of the statistics used to analyze the data. Please, add a section in the methods to explain. Further, in figures 1 and 2 there is no p-value or indication of statistical differences between pH conditions and pre-activation. Please, include those statistics on the results.

(Response) As suggested by the reviewer, a description of statistical analysis has been added to the figures and materials with the exception of Fig. 1G. Fig. 1G is not a repeated experiment, so no statistical data are available.

  • The methods lack important information to understand the experiment and experimental design. Please explain better the treatments used and experiments performed. Also explain better the following:
  • 2. Expand the methods about how the fertilization was performed (media, culture time, time when 2-cell embryos were evaluated, etc).

(Response) The descriptions have been added in Materials and Methods, as suggested by the reviewer.

  • 3. Is it possible to use some standards to extrapolate the exact pHi of eggs, instead of just observing an increase or decrease? How many eggs were assessed per experiment?

(Response) Regarding the estimation of intracellular pH, we had not included it because there are very few examples, but we estimate it to be around 6.3-6.5 when the external pH is 5.5 and 7.0-7.3 when the external pH is 8.2. This result has been added as supplemental data.

  • 4. Expand these methods and specify the different experiments performed. Was SAAF used for assessing motility in the first experiment or just for chemotaxis? What kind of observation chamber was used? How many sperm were evaluated per treatment? How was LECI calculated? Also, explain the methods for the experiment about chemotaxis towards eggs (whole and punctured)?

(Response) We generally followed the reviewer's suggestions and added the necessary information.
SAAF is used only for chemotaxis assays, and the preactivation of sperm motility is done by the high-pH treatment. In original submission, we already mentioned about SAAF only in the chemotaxis assay section of Materials and Methods
The method for evaluating sperm chemotaxis and calculating LECI have already been described in Materials and Methods. Since the method is exactly the same as that in previous papers and this manuscript is limited in space, we have described the description as a citation of the paper.

  • 5. How was the number of sperm in the perivitelline space per egg calculated? How many eggs were evaluated?

(Response) We are verry sorry for wrong description of our calculations. We deeply apologize it. Revised calculations are as follows:

The total number of sperm penetrating the perivitelline space of the egg was 364 in 481 sections of the egg inseminated at pH 8.2 (control conditions). A fixed egg consisted of an average of 18.3 sections (= diameter of a fixed egg: 110 µm / thickness of section: 6 µm) . The average number of sperm in one egg section was 0.76 sperm ( = total number of finding sperm: 364 sperm / total number of sections: 481 sections). Thus, the number of penetrated sperm per an egg was 13.9 ( = 0.76 sperm/section x 18.3 sections/egg).

We have revised the part in Results.

We did not count the number of eggs. There were 481 egg sections, so it is equivalent to 26.3 eggs ( = 481 sections / 18.3 sections/egg) as describing results.

  • The introduction fails to give enough background to understand the present study. Specifically it does not state clearly what is the lack of knowledge that the study wants to fill, what is the aim and why it is important to study it. Further, I would recommend starting the introduction with a brief explanation of general characteristics of the species and their reproduction, as there may be some readers interested in fertilization that are not familiar with the species.

(Response) Thank you very much for your useful suggestions. We have added the sentences as suggested by the reviewer.

  • The discussion is hard to read through. I would recommend organizing it better for discussing the main findings. Moreover, it should be further discussed how pH can affect fertility beyond the main finding of the study (as other effects could be happening at the same time). Last, it would be relevant to mention how other characteristics of seawater beyond pH (for instance osmolality) could be important for triggering fertility.

(Response) In accordance with the reviewer's remarks, we have revised the Discussion section with reference to the other comments of the reviewer.

MINOR CONCERNS

-          Abstract: Please, add aims and conclusions in the abstract. Also, make it clear when referring to past and present study.

We are very sorry for the poor writing. We have revised Abstract with reference to the reviewers' comments.

-          L. 17: Change Fertilized for Fertilize

We have refined the texts of Abstract including the point.

-          L. 18-21: Is this in previous studies? Please specify.

As described above, we have revised Abstract.

-          L. 21: Before <<The pH values>> add <<In the present study,>>.

We have refined the texts of Abstract including the point.

-          L. 23: Change remained for was.

We have corrected it as the reviewer indicated.

-          L. 27: Change triggers for enables. The pH itself does not fertilize the egg, but allows for fertilization to take place).

We have corrected it as the reviewer indicated.

-          L. 27: Before <<An increase in>> add <<We also found that>>.

We have added the phrase as the reviewer indicated.

-          L. 32-36: Instead of this, start with something more specific to the species studied.

Thank you very much for your suggestion. We have revised the initial paragraph of Introduction.

-          L. 42: Change in oviduct to in the oviduct.

We have added the word as the reviewer indicated.

-          L. 58-60: Rephrase to <<The K+ channel Slo3 is also a sperm-specific channel that is regulated by pHi and that may regulate sperm function through changes in pH and membrane potential>>.

We have corrected the sentence as the reviewer indicated.

-          L. 69-70: Erase <<because the molecular mechanisms involved in oocyte maturation and fertilization are complicated>>.

We have erased the sentence as the reviewer indicated.

-          L 71: Change were for are.

We have corrected the word as the reviewer indicated.

-          L. 72: Change were for are.

We have corrected the word as the reviewer indicated.

-          L. 71-77: Add references.

We agree that a bibliography should be cited, but the phenomenon is so well known that we could not find a reference that mention it. If there is a description, it is probably before 1950, so an online search is not possible.

-          L. 79-81: The oocyte maturation process was not studied. Please change this sentence to something that refers to fertilization.

We did not use "maturation" in the sense of oocyte maturation, but in the sense of the last step that makes fertilization possible. We have replaced this word with “preparation of fertilization”, as it was misleading.

-          L. 83: Change this title to <<pH in different ascidian organs and fluids is acidic and lower than sea water>>. Keep low is not correct because it was not tested if the pH was maintained in different conditions and low is relative.

We have corrected the title as the reviewer indicated.

-          L. 88: Did the authors also study Ascidia sydneiensis? Where are those results?

We have some data on A. sydneiensis, but we did not perform all the experiments and there are not many examples. Therefore, we removed them when we were making up the manuscript, but they were left behind in the process. We are very sorry for this, and have removed it.

-          L. 90-92: This sentence is repetitive of the previous lines. Please summarize better the results in Part 2.1. 

We have revised the sentence in accordance with the reviewers suggestion.

-          L. 92-93: Change pH is maintained at an acidic level in not only for pH is acidic not only. 

We have corrected the sentence as the reviewer indicated.

-          L. 94: Change remain for are kept.

We have corrected the word as the reviewer indicated.

-          Table 1: Change pH in the ascidian tissues for pH in ascidian tissues and fluids

We have edited the title of table1 as the reviewer indicated.

-          Table 1: Remove the * in bot the the table and the footnote.

We have deleted the asterisks as the reviewer suggested.

-          L. 99: Change Perivitelline space pH of Phallusia eggs for The pH of the perivitelline space We have corrected the phrase as the reviewer indicated.

-          L.104-107: Add reference

It is our result in the study. We have included a few more details so that the reader can identify it as our result. Since no fertilization occurs, there is no way to write more data.

-          L. 113: change by dilution to by contact.

We have corrected the word as the reviewer indicated.

-          L. 116: Define what SAAF is.

We have added the definition of SAAF as the reviewer indicated.

-          L. 116-117: Where is this data shown? In the corresponding figure, I do not see results of the number of sperm activated in the different pH. Please include this data. Also define what activated means. Do sperm need to show a specific type of motility or any kind of motility to be considered activated? Change were small for was low. Add a dot after Phallusia and start a new sentence. 

This statement is indeed a result, but we have not taken the data properly, partly because it has been already mentioned in previous papers. Thus, this case we have inserted citation at the descriptions in revised manuscript. In addition, the paragraph has been revised for consistency.

-          L. 118-119: Change sentence for <<At pH 8.2 the number of activated spermatozoa was also low even though the activated spermatozoa showed almost the same velocity as that in seawater>>.  

The text has been revised in relation to the indicated points above.

-          L. 121: What did the pre-activation consist of? Cultured in which conditions and for how long?

We have added the section “4.2. Preactivation of sperm motility” in Materials and Methods.

-          L. 124: Add a comma after conditions.

We have corrected it as the reviewer indicated.

-          L. 126: the perivitelline space.

We have corrected it as the reviewer indicated.

-          L. 136: change the eggs inseminating for the number of eggs inseminated. Please specify how insemination was assessed (the number of 2-cell embryos?).

We have corrected the caption, even though we did not added the method of assessment because we thought it is not required in the caption.

-          L. 143: Change entering into for inside. Erase <<(see Materials and Methods)>>.

We have corrected the part as the reviewer indicated.

-          L. 144-149: I believe the way this data is presented is confusing. I suggest explaining this in the methods and for the figure simply present the number of sperm in perivitelline space per oocyte. Also, include error bars in this figure.

We understand your comment that this section is difficult to understand. However, after many revisions, we have settled on this form. It is difficult to write it in a way that is easy to understand in Method. Therefore, we are very sorry, but we would like to leave it as it is. We have revised the text to make it a little easier to understand.

Also, since this data is an aggregate, statistical processing is not available.

-          L. 160: Erase the comma.

We have corrected it as the reviewer indicated.

-          L. 163-164: A suppression of egg fertility is a strong statement. Better erase this sentence as there is already a conclusion of this result part in L. 168.

We have deleted the sentence as the reviewer indicated.

-          L. 165: Why was the preincubation time variable?

Because the egg activation process by pH takes time, the data were taken by varying the time. The data was variable because it was handled by collecting data that yielded approximately the same results.

-          L.171: Define BCECF-.

Spelling out of BCECF was added to Method section 4.4. Although the first appearance is here, we do not consider it appropriate to write it here.

-          L. 179: Erase <<resulting in fertilization>> which is not evaluated in this specific part and in any case the increase of pHi does not induce fertilization but egg fertility.

We have change the phrase ”resulting in fertilization” for “resulting in increase of egg fertility”.

-          L. 183: Was the preincubation 2 h, as mentioned here, or 4 h, as mentioned in the text above?

We are very sorry for misdescription: it is 4 h. We have corrected it.

-          L. 183-184: Open marks refer to non-preincubated eggs. Correct this sentence accordingly.

This is more accurate in the text and inaccurate in the figure explanation. The figure has been corrected.

-          L. 193: Change <<and that at pH 6.5 had little activity>> for <<and that at pH 6.5 further decreased sperm chemotaxis>>.

We have edited the part as the reviewer indicated.

-          L. 195: change furthermore for similarly.

We have changed the word as the reviewer indicated.

-          L. 198: add <<to artificially release the sperm chemoattractants>>. 

We have added the part as the reviewer indicated.

-          L.204: change in the eggs for of the eggs

We have changed the word as the reviewer indicated.

-          L. 206: Erase the comma and <<(see Materials and Methods)>>.

We have changed the part as the reviewer indicated.

-          L. 206-207: Change <<on chemotactic behavior of the sperm>> for <<on the chemotactic behavior of the sperm of C. intestinalis>>. 

We have changed the part as the reviewer indicated.

-          L. 207: Add <<were pre-activated with medium at different pH>>.                                      

We have changed the part as the reviewer indicated.

-          L.207-208: What is the sperm attractant of C. intestinalis and was this also used at the same time than SAAF or was it a separate experiment?

We are very sorry for confusing descriptions: maybe we did not aware that it was fixed during the English proofreading process.

-          L. 208-209: Change <<Activity of the sperm attraction of intact and lysed egg>> to <<Sperm-attracting activity of intact and lysed eggs>>.

We have changed the part as the reviewer indicated.

-          L. 234: Change Normally for In general.

We have changed the part as the reviewer indicated.

-          L.243: This claim is inaccurate. Your results also indicate that once the pH is up it does not come back down when exposed to acidic medium. Therefore there is some kind of pH regulation occurring or the pHi of eggs does not completely depend on ambient pH. 

We have revised the part in accordance with the reviewer’s criticism.

-          L. 245: Specify the enzyme.

In the cited paper, the enzyme has no name. Anyway, it is described as ï½”rypsin-like enzyme.

-          L. 246: Change Furthermore for In our study.

We have changed the part as the reviewer indicated.

-          L.213-218: This first paragraph is confusing and inaccurate as low pH does not trigger egg fertilization but enables it. Moreover, the chemotaxis towards SAAF was affected at pH 6 and the number of activated sperm was also affected by pH, so sperm motility is affected by pH. I would suggest rephrasing to: In this study, it was revealed that the gonoductal gametes of ascidian species are maintained under low-pH which keeps the eggs infertile, and that spawning to seawater increases the ambient pH enabling fertilization. Moreover the study showed that the increase in ambient pH is necessary for sperm chemotaxis as it triggers the release of sperm attractants from the egg. 

We have revised the paragraph in accordance with the reviewer’s suggestion.

-          L. 220: Erase <<which is much higher than that in body>>.

We have deleted the part as the reviewer indicated.

-          L. 224: change the vacuoles for their vacuoles.

We have changed the word as the reviewer indicated.

-          L. 225-226: This is confusing. If pH is acidic in all the body parts, not just the blood, better to erase this hypothesis of the pH being theoretically higher. 

We have deleted the sentence as the reviewer indicated.

-          L.250: may also prevent. 

We have added the word as the reviewer indicated.

-          L. 251: Change However for On the other hand

We have changed the part as the reviewer indicated.

-          L. 257-259: Rephrase to <<However, in ascidian eggs stored in the oviduct, meiosis is arrested at meiotic metaphase I and this stage is preserved after spawning until fertilization>>.

We have changed the sentence as the reviewer indicated.

-          L. 259-261: Rephrase to <<Therefore, our results suggest that other oocyte maturation processes important for fertility may be affected by ambient pH>>.

We have changed the sentence as the reviewer indicated.

-          L.263: Change <<was not observed>> t<<is not present>>.

We have changed the part as the reviewer indicated.

-          L. 266: change <<in male body>> to<<inside the male body>>.

We have changed the phrase as the reviewer indicated.

-          L. 269-271: This statement is inaccurate because your results indicated differences in the number of active sperm, and lower chemotaxis at pH 6.

Based on the reviewers suggestion, we have corrected the description to make it accurate.

-          L.272: What does almost means? Either there was fertilization or not.

We have revised the sentence.

-          L. 273-275: Based on your results, this statement is not correct.

We have revised the part in accordance with the reviewer’s criticism.

-          L. 279: Change <<when the eggs were disrupted>> to <<when the eggs were punctured to mechanically release SAAF>>.

We have changed the part as the reviewer indicated.

-          L. 280-281: This is a hypothesis. The SAAF transporters were not investigated. Instead, mention that the pH affects SAAF release. And if the transporters are mentioned, give more information about how they are involved in SAAF release and include references.

We have revised the sentence in accordance with the reviewer’s suggesion.

-          L. 290: What temperature was the sperm kept at? What is measuring medium, is it the treatment medium? Please rephrase here and in the rest of the methods, and specify that treatments consist of different pH.

Although temperature for keeping sperm have been already described in the original manuscript, we have change the word ‘refrigerated’ to ‘4 °C’.  

As suggested by the reviewer, ï½—e added a description of the composition of the solutions so that it defines "measuring medium".

-          L. 291: Change <<In few experiments, vitelline coat>> to <<In some experiments, the vitelline coat>>

We have changed the part as the reviewer indicated.

-          L. 292: Change glade for blade.

We have changed the part as the reviewer indicated.

-          L. 291-293: Include that these eggs are called dechorionated as in the Results section.

We have changed the part as the reviewer indicated.

-          L. 327: Change measurements for treatments.

We have changed the part as the reviewer indicated.

-          L. 343. The number.

We have changed the part as the reviewer indicated.

-          L. 344. the number.

We have changed the part as the reviewer indicated.

-          L. 347. Change <<by spawning>> to <<by contact with seawater after spawning>>.

We have inserted the phrase as the reviewer indicated.

-          L. 349 -350: Rephrase this sentence.

We have revised Conclusion section in accordance with the reviewer’s suggestion.

-          L.351-353: In which species?

We have revised Conclusion section in accordance with the reviewer’s suggestion.

-          L. 354: Change study to understand.

We have replaced the word as the reviewer indicated.

Reviewer 2 Report

This is elegant study that shows pH changes at spawning which play an important role in egg fertility. The big plus of the study is experimental model Phallusia philippinensis.  The study is clearly written and well-designed. Figures are presented in detail and support data. No concerns are found. The manuscript should be published at present form.

If the Editors require permission number for the study and information on number of used animals and their age -it should be provided.

Author Response

This is elegant study that shows pH changes at spawning which play an important role in egg fertility. The big plus of the study is experimental model Phallusia philippinensis.  The study is clearly written and well-designed. Figures are presented in detail and support data. No concerns are found. The manuscript should be published at present form. 

If the Editors require permission number for the study and information on number of used animals and their age -it should be provided.

(response)

We appreciate your favorable comments on this manuscript.

We have made some revisions in response to the other reviewer's comments.

Ascidians are invertebrates and are not subject to regulation under animal welfare guidelines in Japan.